# A Computational Analysis of the Influence of a Pressure Wire in Evaluating Coronary Stenosis

**Jie Yi** [1] , **Fang-Bao Tian** [2], **Anne Simmons** [1] **and Tracie Barber** [1,*]

1   School of Mechanical & Manufacturing Engineering, University of New South Wales,
    Sydney, NSW 2052, Australia; jie.yi@unsw.edu.au (J.Y.); a.simmons@unsw.edu.au (A.S.)
2   School of Engineering & Information Technology, University of New South Wales,
    Canberra, ACT 2612, Australia; fangbao.tian@adfa.edu.au
*   Correspondence: t.barber@unsw.edu.au

**Abstract:** Cardiovascular disease is one of the world's leading causes of morbidity and mortality. Fractional flow reserve (FFR) was proposed in the 1990s to more accurately evaluate the functional severity of intermediate coronary stenosis, and it is currently the gold standard in cardiac catheterization laboratories where coronary pressure and flow are routinely obtained. The clinical measurement of FFR relies on a pressure wire for the recording of pressures; however, in computational fluid dynamics studies, an FFR is frequently predicted using a wire-absent model. We aim to investigate the influence of the physical presence of a 0.014-inch ($\approx$0.36 mm) pressure wire in the calculation of virtual FFR. Ideal and patient-specific models were simulated with the absence and presence of a pressure wire. The computed FFR reduced from 0.96 to 0.93 after inserting a wire in a 3-mm non-stenosed (pipe) ideal model. In mild stenotic cases, the difference in FFR between the wire-absent and wire-included models was slight. The overestimation in severe case was large but is of less clinical significance because, in practice, this tight lesion does not require sophisticated measurement to be considered critical. However, an absence of the pressure wire in simulations could contribute to an over-evaluation for an intermediate coronary stenosis.

**Keywords:** pressure wire; fractional flow reserve; coronary stenosis; computational fluid dynamics

## 1. Introduction

The risk of arterial stenosis is a large health care burden worldwide. In the investigation of atherothrombosis-related events, the REACH (REduction of Atherothrombosis for Continued Health) Registry recruited approximately 68,000 patients from 44 countries across six regions. Statistical analysis illustrated that as many as 30% of coronary artery disease (CAD) patients had systemic atherosclerotic disease (disease in more than one arterial bed) [1]. Clinically, both routine diagnosis using medical imaging techniques and invasive functional assessment procedures are used for stenosis identification in coronary arteries.

Fractional flow reserve (FFR) is one of the most notable indices to estimate the functional severity of ischemia-inducible coronary stenosis [2] and is considered the gold standard in cardiac catheterization laboratories where coronary pressure and flow are routinely obtained. The FAME (Fractional flow reserve versus Angiography for Multivessel Evaluation) study demonstrated that percutaneous coronary intervention (PCI) guided by FFR significantly improved outcomes in one year in patients compared with PCI guided by angiography alone [3]. In patients with stable CAD and functionally significant stenosis (FFR $\leq$ 0.8), FFR-guided PCI plus the best available medical therapy were also associated with a decreased rate of 9.5% of urgent revascularization [4].

FFR is defined as the ratio of the maximal myocardial blood flow supplied by the target coronary artery in the presence of stenosis ($Q_s^{max}$) to the maximal myocardial blood flow in that same territory in the hypothetical case that the stenosis were removed and the epicardial vessel were completely normal ($Q_n^{max}$) [5]. In hyperemia conditions, the

resistance is assumed to be constant and equivalent. The venous pressure is small and could be negligible. FFR therefore equals the distal pressure ($P_d$) divided by the aortic pressure ($P_a$).

Clinically, FFR is measured invasively in intermediate stenosis cases, after vasodilator stimulus. A sensor mounted on the guiding catheter is used to record the value of $P_a$ in the coronary ostium, and a 0.014-inch ($\approx$0.36 mm) pressure wire is introduced to record the value of $P_d$ at the distal end of a stenosis [6].

In order to reduce or eliminate the invasive procedures to obtain the value of FFR, a number of computer-related methods have been investigated [7–11]. FFR derived from coronary tomography angiography ($FFR_{CTA}$) was predicted via computational fluid dynamics (CFD) study (HeartFlow, Inc., Redwood City, CA, USA) [7]. The lumped-parameter coronary artery model was prescribed at the outlet for an unsteady simulation. A high diagnostic performance of $FFR_{CTA}$ was concluded compared with invasively measured FFR [12]. FFR presents a proportion of the averaged flow or pressure over the cardiac cycle. Virtual FFR computed from steady state was proposed and compared with invasively measured data in 21 patients [11]. It was found that the computational burden reduced to 1/16 in comparison to unsteady flow simulation and the accuracy maintained to be high. The non-CFD method of angiography derived FFR ($FFR_{angio}$) was generated according to rapid flow analysis (CathWorks Ltd., Kefar Sava, Israel), and a high concordance between $FFR_{angio}$ and wire-based FFR was concluded [10]. Although good agreements are indicated in the previous studies, one major inconsistency between clinically measured FFR and numerically predicted FFR is the presence of the pressure wire.

A wire is usually considered so small that its impact is negligible. However, in a study on the applicability of a 0.015-inch fluid-filled guide wire, the presence of the guide wire led to an overestimation (>20%) in cases of severe stenosis (>90% area reduction) [13]. An in vitro experiment conducted by Koustubh et al. also showed that flow obstruction was observed with an increased trans-stenotic pressure drop due to the guidewire insertion (diameter of 0.35 mm) [14]. To further account for its impact, a combined anatomical and functional index of Lesion Flow Coefficient was proposed for the detection of CAD by the same research group [15].

In this paper, we investigate the influence of the physical presence of a pressure wire, using CFD simulations. Both ideal geometries of healthy and stenotic models (30–70% diameter reduction), and a patient-specific model have been studied. The flow rates and pressure have been analyzed. The values of virtual FFR are compared in the wire-absent and wire-included models, and the effect of a wire on evaluating coronary stenosis has been identified.

## 2. Methods

### 2.1. Modeling Geometry

Ideal axisymmetric geometries with polar coordinate (r, z) were created (Figure 1). Three-dimensional models were generated by revolving around the *z*-axis, including a non-stenosed (pipe) case and five stenotic cases with 30 to 70 percent diameter stenosis (%DS) in the increment of 10%. The profile for stenotic models is the same as described in the study of Siouffi et al. [16]. The diameter (D) of idealized models is 3 mm, which is typical for a coronary artery [17]. The lesion length is 4 times D, and the total length is 40 times D. A 0.014-inch ($D_{wire}$ = 0.36 mm) pressure wire was inserted and positioned 10 times D distal to the minimal lumen area (MLA) [18]. The value of $P_d$ is captured at the central point at the tip of pressure wire. FFR is calculated as the pressure ratio (Equation (1)):

$$FFR = \frac{P_d}{P_a} \tag{1}$$

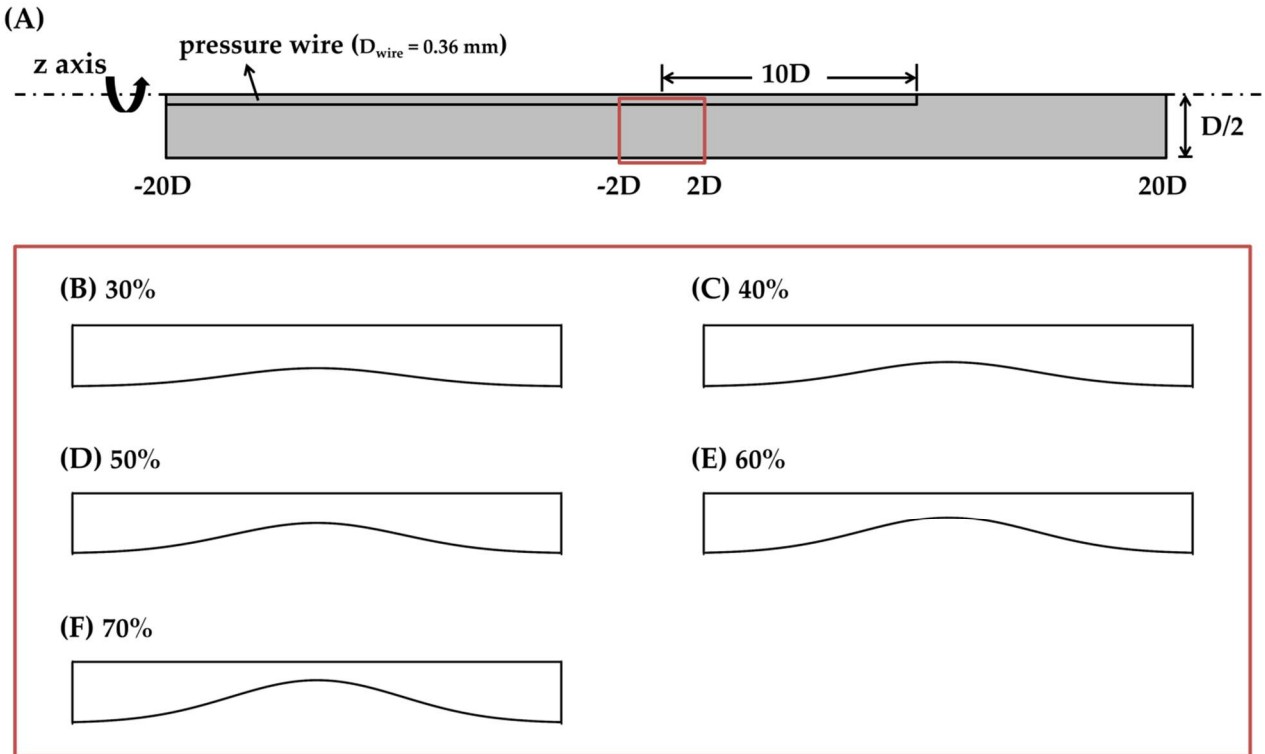

**Figure 1.** A schematic illustration of ideal models inserted with a pressure wire. (**B**–**F**) are zoom-in views of stenosis locating at −2D to 2D in the coordinate. (**A**) represents the non-stenosed (pipe) case, while (**B**–**F**) represent the stenotic domains of 30, 40, 50, 60, 70% DS models, respectively.

A patient-specific model of left anterior descending coronary artery has been reconstructed as well from intravascular ultrasound images, which were obtained in catheterization laboratory at 15 frames per second (Axiom Artis, Siemens, Germany) (Figure 2). The diameter at the inlet is 2.97 mm, while the diameter at the outlet is 1.15 mm. The minimal lumen diameter at the stenosis throat is 0.65 mm. The area at the narrowing region is about 0.57 mm$^2$, and the area reduction of this case is 79% (DS% of about 54%). The lesion length is 13.14 mm, and the total length is 63.24 mm. The pressure wire was introduced along the centerline in the position of 2 cm distal to the narrowest area. The value of $P_d$ is captured at the central point at the tip of pressure wire, and FFR is calculated by Equation (1).

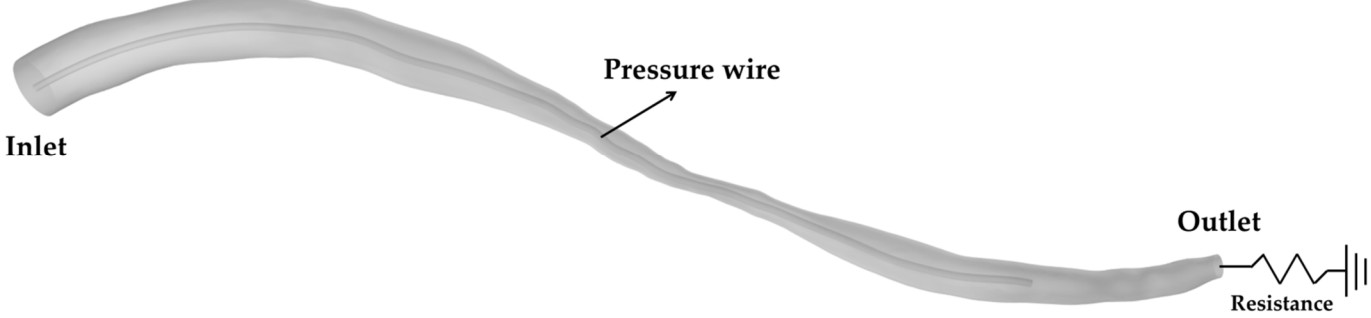

**Figure 2.** A patient-specific model with a 79% area reduction inserted with a pressure wire along the centerline. The resistance prescribed at the outlet denotes the downstream mircocirculatory resistance.

### 2.2. Numerical Assumptions and Boundary Condition

COMSOL Multiphysics 5.3 has been employed for the CFD simulations. The blood is assumed to be an incompressible and Newtonian fluid with a density of 1050 kg/m$^3$ and a viscosity of 0.0035 Pa·s [19,20]. In practice, the vasodilator is injected to maintain the maximal hyperemic condition and the vessels expand to the maximum, vessels are

therefore considered as rigid conduits with a non-slip wall boundary in the simulation. A mean blood pressure of 93 mmHg was applied at the inlet for the steady flow simulation, which denotes the value of $P_a$ in our study. Based on the load independence principle of FFR [21], a resistance boundary is considered to represent the downstream arterial system [22]. The pressure in the artery is a function of the flow rate and resistance. At the outlet, a user-defined function of the resistance was applied, which couples the blood pressure and flow rate. The value of resistance in maximal hyperemia reduced to 0.24 of that in resting condition from the clinical data of Wilson et al. [23]. In this study, the resistance of the downstream microcirculatory was set as $4 \times 10^9$ Pa·s/m$^3$ in the predicting models according to the published data [22]. The relative tolerance was set to be $10^{-4}$ for the converged results.

### 2.3. Navier–Stokes Equation

In the CFD studies, cardiovascular blood flow was modeled using the Navier–Stokes equations (Equation (2)):

$$\nabla \cdot u = 0 u \cdot \nabla u = -\frac{\nabla p}{\varrho} + \frac{\mu}{\varrho}\nabla^2 u \tag{2}$$

where u is the fluid velocity, p is the pressure, $\varrho$ is the density, and $\mu$ is the dynamic viscosity. Finite element method has been performed to solve these equations for its good fidelity and robustness. The outlet condition is coupled to the inlet condition by an iteration method that updates the pressure and velocity as the iteration processes. The coupling calculations continue until the velocity and pressure converges, using a segregated solution approach.

### 2.4. Grid Independence Study

To determine the effect of the mesh on the solution accuracy, two different meshes in ideal models were created, containing $5 \times 10^4$ and $2 \times 10^5$ elements, respectively. The minimum element quality was greater than 0.5 in each case. For the patient-specific model, 1.5 million- and 3 million-element meshes were generated. The parameter of $P_d$ is sensitive and significant in our study, and it is selected to determine an appropriate element size. The difference in the value of $P_d$ generated by the normal and fine grids was observed to be less than 0.1% for the idealized models and patient-specific case alike. Hence, the $5 \times 10^4$ and 1.5 million element grids were deemed to be appropriate to use in the remaining ideal and patient-specific model studies, respectively.

## 3. Results and Discussion

### 3.1. Blockage Ratio and Flow Obstructive Effect

A physical comparison between the size of the pressure wire and of the vessel is present via analyzing the blockage ratio (Table 1). The blockage ratio is equal to the area of wire ($A_{wire}$) divided by the MLA (Equation (3)):

$$\text{Blockage Ratio} = \frac{A_{wire}}{MLA} \tag{3}$$

**Table 1.** The blockage ratio and flow obstructive rate for both ideal and patient-specific models.

| Model | DS% | Blockage Ratio | Flow Obstructive Rate |
|---|---|---|---|
| Ideal | 0% | 1.4% | 3.4% |
| | 30% | 3.0% | 3.8% |
| | 40% | 4.0% | 4.5% |
| | 50% | 5.8% | 5.8% |
| | 60% | 9.0% | 9.8% |
| | 70% | 16.0% | 20.3% |
| Patient-specific | 54% | 17.8% | 18.7% |

With the growth of the narrowing, the blockage ratio increases from 1.4% to 16.0% nonlinearly in the idealized models. The pressure wire has a greater obstructive effect physically in the severe case than in the mild cases. In the patient specific case, although its diameter reduction is smaller compared with a 60% ideal stenotic model, the physical blockage effect is larger (17.8%).

The obstructive effect of the pressure wire accounts for the decrease of the flow rate after its insertion in the predicting models. The flow obstructive rates are calculated and shown in Table 1 according to the simulated flow rates obtained from both wire-absent and wire-included models. In mild stenotic cases (30, 40, 50% DS), the rate of the flow obstruction remains within 6%, while it increases greatly to 20% in the ideal severe case (70% DS). In the real situation, vessels are normally tortuous and eccentric. Therefore, when a pressure wire is introduced, the blockage rate becomes higher in practice than in the ideal model with a similar degree of stenosis, and the flow obstructive rate is even higher in a 54% DS patient specific case than in a 60% ideal stenotic case in our study (18.7% vs. 9.8%).

### 3.2. Pressure

Pressure contours for all ideal and patient-specific models are present in Figures 3 and 4. In order to visualize the variations more easily, different legends in the different models are utilized. An apparent pressure drop is observed in the downstream region compared between every two paired models. With the increasing of the severity of the stenosis, the pressure drops larger in the ideal models.

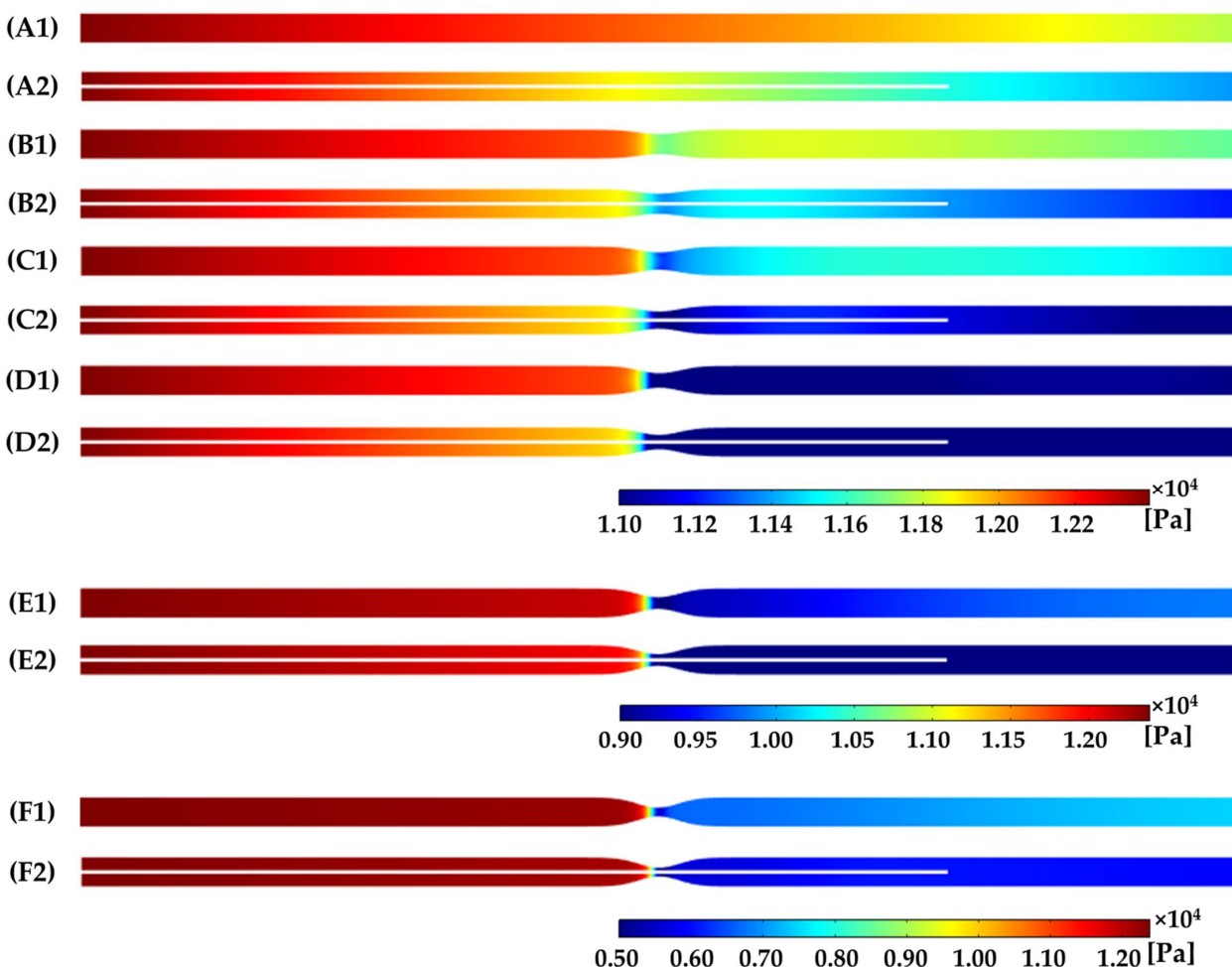

**Figure 3.** Pressure contours of the ideal models in a cross section. (**A**) denotes a healthy model, while (**B**–**F**) denote the stenotic model with 30–70% DS in the increment of 10%, respectively. (**1**) and (**2**) denote the paired model without and with a pressure wire. In order to visualize the variations more easily, different legends are performed in different cases.

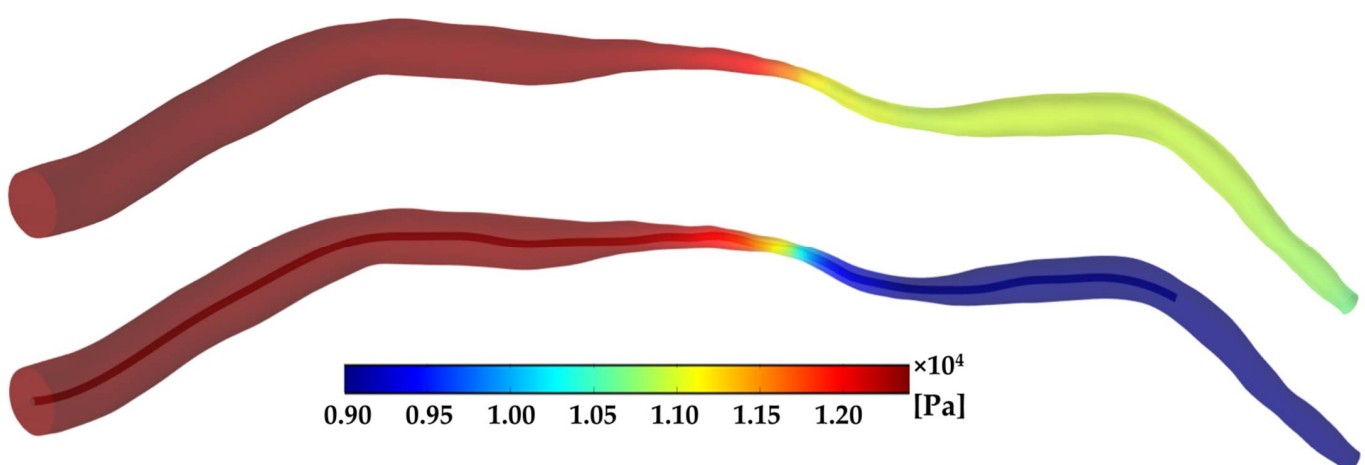

**Figure 4.** Pressure contours for a patient-specific case without and with a pressure wire.

A non-stenosed (pipe) model has been computed as a baseline (Figure 3A). The values of $P_d$ are 11,927 and 11,563 Pa in the models without and with a pressure wire, respectively. In the model with 30% DS, the value of $P_d$ is 11,789 Pa without the presence of the pressure wire, while it drops to 11,379 Pa after the insertion (Figure 3B). In 40% and 50% DS models, the values of Pd decrease slightly as well in wire-absent and wire-included conditions (11,575 vs. 11,129; 11,013 vs. 10,504) (Figure 3C,D). When a stenosis increases to 60%, the value of Pd reduces from 9645 to 8994 Pa after inserting a pressure wire (Figure 3E). In the severe model (70% DS), the pressure drops significantly, and the presence of a wire leads to a decrease of $P_d$ from 7106 to 6032 Pa (Figure 3F). In summary, a growth of the narrowing contributes to the decrease of pressure, and the insertion of pressure wire in the simulations accelerates the (measured) pressure drop.

A patient-specific case with moderate stenosis has also been studied. The value of $P_d$ in the non-wire model is 10,927 Pa, while it decreases to 8855 Pa in the wire-included model, respectively (Figure 4). A larger pressure drop between wire-absent and wire-included models is observed in the patient-specific case compared with the ideal models. There are two causes. Firstly, in the cardiovascular system, the downstream arterial diameter usually becomes smaller even though a stenosis is not present and the individual is healthy, as can be demonstrated in the geometry in Figure 2—the outlet diameter is smaller than the upstream arterial diameter. A reduced diameter downstream of the vessel may accelerate the decrease of the pressure [24]. Secondly, pressure reduces due to the aforementioned flow obstructive effect caused by the pressure wire.

### 3.3. FFR

FFR is calculated by the ratio of the computed $P_d$ to $P_a$. $P_d$ is obtained at the central point at the tip of pressure wire, while $P_a$ is assumed to be 93 mmHg at the inlet in our study. The value of FFR in the non-stenosed (pipe) model is 0.96 when the wire is not inserted, while it dropped by 0.03 due to the insertion of the wire. Figure 5 shows the value of FFR in various ideal stenosis models. In mild stenosis cases (30, 40, 50% DS), the values of FFR are 0.95, 0.93 and 0.89 in the models without a pressure wire, respectively, while they decrease to 0.92, 0.90 and 0.85 in those with a wire. The presence of the pressure wire contributes to a relatively slight overestimation of around 4%. In the stenotic model of 60% DS, the value of FFR reduces from 0.78 to 0.73 in the two comparative models, and it drops significantly in the 70% DS model from 0.57 to 0.49 with an overestimation rate of 15%.

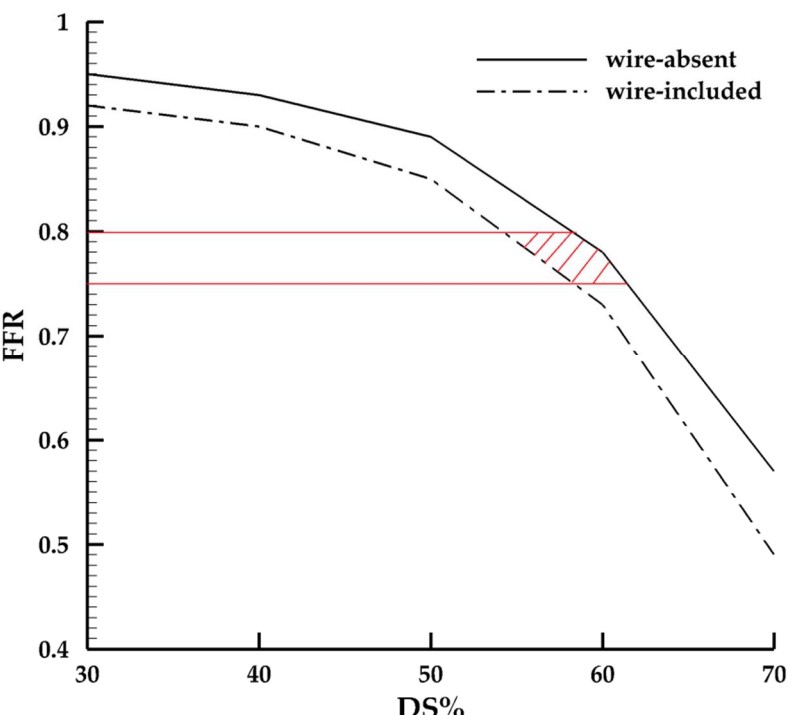

**Figure 5.** The value of FFR in the idealized models with 30% to 70% DS in the increment of 10%. The red region denotes the "gray zone" of FFR from 0.75 to 0.80.

In the patient-specific case, the value of FFR in the wire-absent model is 0.88. It dropped to 0.71 when the pressure wire is introduced, which contributes to a high overestimation as 19%.

Based on the results, introducing a pressure wire in CFD simulations only has a minor influence on the FFR value for mild stenotic cases, and therefore a negligible impact on the stenosis evaluation and diagnosis. A large overestimation of the functional severity of stenosis was observed in the severe case; however, this is of less significance in clinical practice because such a tight lesion does not require sophisticated measurements to be considered critical and the intervention may be performed after routine diagnosis rather than invasive functional assessment [25]. This can be further illustrated in the 70% DS ideal case in our study, a value of FFR of 0.57 has already indicated a severe condition. Although the insertion of the pressure wire leads to a lower value of FFR of 0.49, the impact is of less significance for clinical guidance.

The cut-off value of FFR is 0.75 based on the clinical statistics analysis [26], yet cardiologists may elect to perform PCI when FFR detected is between 0.75 to 0.80 if the clinical scenario suggests ischemia [27]. In the intermediate stenosis from 50 to 60% ideal stenotic cases, FFR predicted may be greater than 0.8 in a wire-absent model, and it may drop to the "gray zone" if the impact of the pressure wire is considered in the CFD study, as is showed near the red region in Figure 5. That makes it problematic for clinical decisions and may result in a false evaluation when the value of FFR is non-invasively predicted. Furthermore, in the patient-specific intermediate case, a value of FFR as 0.88 certainly indicates a non-significant ischemia clinically. However, the value of 0.71 in the simulated model with a pressure wire inserted indicates a functionally significant stenosis, and further intervention may be considered. While uncertainties resulted from the current simulation strategies [28] may lead to a deviation of the virtual FFR compared with the physiological situation, the difference of FFR in patient-specific models reveals that the pressure wire has an overestimation impact on evaluating intermediate coronary artery in CFD studies.

## 4. Limitations

The pressure wire is assumed with little deformation and rigidity in this research. However, the wire in practice is not static and moves with the unsteady blood flow. Further studies of the relationship between the movement of the pressure wire and the fluid flow need to be investigated via fluid–structure interaction analysis.

In addition, only one patient-specific model has been studied in our research. In order to unmask the real influence of the pressure wire, more patient-specific cases need to be included in the future.

## 5. Conclusions

A pressure wire is usually considered to be small, and its impact is neglected in most CFD studies. In our research, both ideal and patient-specific models have been simulated with the absence and presence of a pressure wire. In conclusion, an absence of the pressure wire in FFR-predicting models could result in an over-evaluation of the coronary stenosis, and the influence needs to be given serious consideration when FFR is close to the "gray zone" (0.75–0.80).

**Author Contributions:** Conceptualization, T.B. and J.Y.; methodology, J.Y.; software, J.Y.; validation, J.Y.; formal analysis, J.Y. and T.B.; investigation, J.Y.; resources, T.B.; data curation, J.Y.; writing—original draft preparation, J.Y.; writing—review and editing, T.B. and F.-B.T.; visualization, J.Y.; supervision, T.B., F.-B.T and A.S.; project administration, T.B.; funding acquisition, T.B. All authors have read and agreed to the published version of the manuscript.

**Funding:** This research was funded by the Australia Research Council, grant number LP150100574.

**Institutional Review Board Statement:** Not applicable.

**Informed Consent Statement:** Not applicable.

**Data Availability Statement:** The data presented in this study are available on request from the corresponding author.

**Conflicts of Interest:** The authors declare no conflict of interest.

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
