# Peer review of "A Computational Analysis of the Influence of a Pressure Wire in Evaluating Coronary Stenosis"

_fluids, doi:10.3390/fluids6040165_

Round 1
Reviewer 1 Report
The authors of the original article A computational analysis of the influence of a PressureWire in evaluating coronary stenosis present an interesting and potentially clinically important findings. The computed FFR reduced from 0.96 to 0.93 after inserting a wire in a 3-mm non-stenosed (pipe) ideal model. Though difference in FFR between the wire-absent and wire-included models was slight in mild stenotic cases, an absence of the pressure wire in simulations could contribute to an over-evaluation for an intermediate coronary stenosis.
The article is well prepared and a good quality of the English.
I have only some minor issues and questions to the authors:
- The pressure sensor for the aortic pressure is outside the the guiding catheter that is filled with saline.
- FFR based on the PressureWire usage during maximal hyperemia has served as the gold standard for functional lesion evaluation. Thus, the differences in the pressures with/without the wire should be clearly dedicated to the non-wired indices and techniques, i.e. Discussion should be expanded.
- How may the pipe model help the clinicians in angulated arteries?
- What should be the smallest vessel diameter for wired-FFR measurement?
Reviewer 2 Report
I consider your manuscript to be of high quality.
Author Response
Thank you for your comments.
Reviewer 3 Report
Review Report fluids-157112
by Yi et al.
The authors present a CFD study with the aim of analysing the influence of the pressure wire in evaluating coronary stenosis by means of FFR. Studies in the literature do not consider the wire. The authors present a number of idealized models and one patient-specific geometry of a stenotic artery in the presence and in the absence of the pressure wire. The results of the computational simulations reveal a low effect of the wire even its physical presence inside the artery causes an overestimation of the FFR. The authors state that the latter can be important in the case an intermediate stenosis.
GENERAL COMMENTS
The interest of the study is limited, as the results reveals that actually the wire can be neglected. With all the assumptions made, it is also unclear if for an intermediate stenosis the produced ovestimation of the FFR may really be important. The authors should use only patient specific models for obtaining results of FFR which can be really trusted. The comparison between patient specific and idealized models (as the authors recognize) are quite different. In my opinion, the idealized models are too simplified and should not be used (no tapering, no bifurcations, axysimmetric stenosis...). Also the boundary conditions are quite simplified.
In this view, the scientific soundness of the work and its clinical application is low.
TECHNICAL COMMENTS
I encourage the authors to use adequate models for such important clinical problem. The should probably use only patient specific models and patient specific flow and/or pressure data.
In the Introduction, any formula should be removed and moved to the Materials and Methods or Results section. Also, an extensive bibliography research is needed, as many relevant work are missing.
Information about the patient and the creation of the patient specific model should be given at page 3.
The outlet boundary condition should be accurately explained.
Why the flow is not applied to the inlet? this can be non invasively measured. The authors applied a constant pressure. However, this is an important assumption. PAtient specific data are necessary for the FFR computation as well as patient specific geometries.
The grid independence study should be moved to the section Materials and Methods, when talking about the mesh. The mesh independence study is not a result.
Why did the authors change the legend of the figure 3? For a better comparison, the same legend should be used.
Lines 182-191: In view of your comments, the results provided by the idealized models should not be completely invalidated? This is one of the reason why the study should be accomplished using patient specific data only.
The application of the results coming from idealized models to the clinics is not possible as the results cannot be trusted.
Round 2
Reviewer 1 Report
The authors addressed the comments accordingly.
Author Response
Thank you for the review of our paper.
Reviewer 3 Report
The authors have answered to all the questions highlighted in the review so that I am happy to recommend the manuscript for publication. My last minor suggestion is to include some additional study regarding the computation of the FFR in the Introduction. Even these studies do not include the wire, it could be important to create the necessary context (around the FFR computation) for the present study.
Author Response
Thank you for the review of our paper. We have carefully considered the suggestions and made some changes in the manuscript.
In line 58-59, the lumped-parameter coronary model used in the published work was mentioned and in line 132-134, citations were added to explain the resistance boundary, which may help the reader be more clear about the boundary condition set-up and the simulation of FFR.
In line 60-64, publications were added about the steady flow simulation of FFR, which may be helpful to understand the computation of virtual FFR.